# Multi-Trait Exome-Wide Association Study of Back Pain-Related Phenotypes

**DOI:** 10.3390/genes14101962

**Published:** 2023-10-19

**Authors:** Irina V. Zorkoltseva, Elizaveta E. Elgaeva, Nadezhda M. Belonogova, Anatoliy V. Kirichenko, Gulnara R. Svishcheva, Maxim B. Freidin, Frances M. K. Williams, Pradeep Suri, Yakov A. Tsepilov, Tatiana I. Axenovich

**Affiliations:** 1Institute of Cytology and Genetics, Siberian Branch of Russian Academy of Sciences, 630090 Novosibirsk, Russia; zor@bionet.nsc.ru (I.V.Z.); elgaeva@bionet.nsc.ru (E.E.E.); belon@bionet.nsc.ru (N.M.B.); kianvl@bionet.nsc.ru (A.V.K.); gulsvi@mail.ru (G.R.S.); tsepilov@bionet.nsc.ru (Y.A.T.); 2Department of Natural Sciences, Novosibirsk State University, 630090 Novosibirsk, Russia; 3Vavilov Institute of General Genetics, Russian Academy of Sciences, 119333 Moscow, Russia; 4Department of Biology, School of Biological and Behavioural Sciences, Queen Mary University of London, London EC1M 6BQ, UK; maxim.freydin@kcl.ac.uk; 5Department of Twin Research and Genetic Epidemiology, King’s College London, London SE1 7EH, UK; frances.williams@kcl.ac.uk; 6Seattle Epidemiologic Research and Information Center, VA Puget Sound Health Care System, Seattle, WA 98108, USA; 7Division of Rehabilitation Care Services, Seattle, WA 98208, USA; 8Clinical Learning, Evidence, and Research Center, University of Washington, Seattle, WA 98195, USA; 9Department of Rehabilitation Medicine, University of Washington, Seattle, WA 98195, USA

**Keywords:** linear combination of traits, chronic back pain, dorsalgia, intervertebral disc disorder, *SLC13A1*, *FSCN3*, shared heritability, loss-of-function (LoF) variant, rare and ultra-rare genetic variants

## Abstract

Back pain (BP) is a major contributor to disability worldwide, with heritability estimated at 40–60%. However, less than half of the heritability is explained by common genetic variants identified by genome-wide association studies. More powerful methods and rare and ultra-rare variant analysis may offer additional insight. This study utilized exome sequencing data from the UK Biobank to perform a multi-trait gene-based association analysis of three BP-related phenotypes: chronic back pain, dorsalgia, and intervertebral disc disorder. We identified the *SLC13A1* gene as a contributor to chronic back pain via loss-of-function (LoF) and missense variants. This gene has been previously detected in two studies. A multi-trait approach uncovered the novel *FSCN3* gene and its impact on back pain through LoF variants. This gene deserves attention because it is only the second gene shown to have an effect on back pain due to LoF variants and represents a promising drug target for back pain therapy.

## 1. Introduction

Back pain (BP) is the leading cause of disability worldwide [1,2]. Several BP-related phenotypes are commonly used for genetic analysis, with chronic back pain (CBP) being the most severe form, defined as self-reported BP lasting more than 3 months, and the most commonly studied to date [3,4,5,6]. Dorsalgia is an electronic health record (EHR)-based phenotype defined by clinical diagnostic codes for back pain and neck pain, but predominantly reflects the former. Intervertebral disc disorder (IDD) is another BP-related phenotype defined by EHR-based codes for intervertebral disc degeneration that are typically (but not necessarily) used when BP is present [7,8]. Several dozen loci associated with CBP, dorsalgia, and IDD have been identified by genome-wide association studies (GWAS) [7,8,9,10]. Genes associated with BP-related phenotypes are involved in cartilage and bone biology, neurological, and inflammatory processes, with some genes being associated with more than one phenotype [8]. In the largest genetic study of BP-related phenotypes, six loci marked by the *COL11A1*, *SPON2*, *C6orf106*, *GSDMC*, *CHST3*, and *FOXA3* genes were shown to affect both IDD and dorsalgia [8]. The *SOX5* and *PANX3* genes have been associated with CBP [9,11] and IDD [8]. In the current report, we use the term “back pain-related phenotypes” to refer collectively to these three different yet related phenotypes associated with back pain.

All BP-related traits are complex heritable traits with narrow-sense heritability estimated from twin studies in the range of 30% to 70% [12,13,14,15]. Liability-scale SNP-based heritability explained by common variants has been estimated at 12% for CBP [16], 11.4% for dorsalgia (https://nealelab.github.io/UKBB_ldsc/h2_summary_M54.html, accessed on 1 November 2022), and 8% for IDD (https://nealelab.github.io/UKBB_ldsc/h2_summary_M51.html, accessed on 1 November 2022). These estimations are substantially lower than the heritability expected from classical twin studies. Several ways can be suggested to explain this missing heritability, such as using rare (minor allele frequency, MAF < 0.01) variants [17] and more powerful methods of association analysis.

Analysis of rare variants in order to estimate their impact on complex traits is now possible due to whole-exome sequencing technology. Exome sequencing data have already been used for the analysis of CBP, and the association with loss-of-function (LoF) variants in the SLC13A1 gene was detected [18].

In this study, we also used the exome sequencing data from the UK biobank project that has been used by Ao et al. (2023). However, our and the cited studies differ in materials and methods. Ao et al. used only one BP-related phenotype, namely, CBP, but we considered three different BP-related phenotypes: CBP, dorsalgia, and IDD. The lack of a standard definition of BP poses a significant challenge in identifying new genetic loci for BP. To address these challenges and enhance our understanding of the genetic influence on BP, we used three BP-related phenotypes. The genetic nature of all these phenotypes is common because the genetic correlations between them are about 90% [6,8].

We applied a new method of multi-trait association analysis [19]. This method combines several genetically correlated traits into a single multi-trait by maximizing its heritability. This approach allows the identification of the common genetic background for several genetically correlated traits and increases the power of association analysis due to the lower genetic heterogeneity of multi-trait in comparison with the original traits. It has been shown that such analysis can identify genes that were undetected by the separate analysis of each original trait [6,19].

In this study, our aim was to analyze the genes responsible for differences in the risk of BP-related phenotypes among individuals. We achieved this by examining rare and ultra-rare variants using exome sequencing data from the UK Biobank. To increase the power of the analysis, we employed a multi-trait approach.

## 2. Materials and Methods

### 2.1. Overview of the Study Design

This study was carried out using the UK Biobank 200k exome dataset (https://biobank.ndph.ox.ac.uk/ukb/field.cgi?id=23156, accessed on 14 November 2021) and phenotypes of three BP-related traits: CBP, dorsalgia, and IDD (https://biobank.ndph.ox.ac.uk/ukb/field.cgi?id=41270, accessed on 3 November 2022).

The study design included three stages. During the first stage, we calculated the summary statistics (z-scores, effect sizes, *p*-values) using an exome-wide association study (EWAS) for three BP-related traits and for a new multi-trait, which condenses the genetic background common for all original traits analyzed. In the second stage, we performed gene-based association analyses of all original traits and multi-trait. In the third stage, we performed a conditional analysis of the identified genes to reduce the influence of strong association signals located outside the gene. 

### 2.2. Data

We used information about 188,237 white participants from the UK Biobank 200k exome dataset. Information on self-reported ethnic background was captured in data field 21000. Three BP-related phenotypes were analyzed: CBP (N = 178,963), dorsalgia (N = 183,652), and IDD (N = 183,652). The characteristics of the patients with the different BP-related phenotypes are presented in Appendix A. The prevalence of three BP-related phenotypes was 17.97% for CBP, 3.48% for dorsalgia, and 1.76% for IDD in a large sample of white people from the UK Biobank (N = 459,161). Many individuals have two or three of these BP-related phenotypes simultaneously (Appendix A). In particular, 68% of individuals with IDD also have CBP and/or dorsalgia.

For CBP, cases and controls were defined based on questionnaire responses (data field 3571 “Back pain for 3+ months”). First, participants responded to “Pain type(s) experienced in the last months”, followed by questions inquiring if the specific pain had been present for more than 3 months. Those who reported back pain lasting more than 3 months were considered chronic back pain cases, while participants reporting no such pain were considered controls. Individuals who preferred not to answer or reported more than 3 months of pain all over the body were excluded from the study.

For dorsalgia and IDD, EHR-based phenotypes were defined by the level 2 ICD-10 codes M54 and M51, respectively. All participants with either of these level 2 medical codes (M54 and M51 and/or codes of a lower level (level 3 and/or 4, which are subsumed under the level 2 code)) were considered as cases. For an overview of ICD-10 codes in the UK Biobank, see https://biobank.ndph.ox.ac.uk/ukb/field.cgi?id=41202 (accessed on 3 November 2022).

We used exome sequencing data from the UK Biobank 200k release. We carried out quality control as advised in https://biobank.ndph.ox.ac.uk/showcase/refer.cgi?id=914 (accessed on 3 November 2022). We used BCFtools 1.16.1 (https://samtools.github.io/bcftools/, accessed on 5 November 2022) to filter single nucleotide variants (SNV) directly on the multi-sample VCF (data field 23156). After quality control, there were 11,375,237 autosomal SNVs with MAF < 0.01.

### 2.3. EWAS of Original Traits

A single-variant association analysis for each BP-related trait was performed using the fastGWA-GLMM tool, version 1.94.0 beta [20]. We conducted the analysis using a sparse genomic relationship matrix reflecting genomic relatedness between individuals to account for random effects mediated by relationships between individuals. This matrix was calculated using fastGWA-GLMM (–make-bK-sparse option with default parameters) for the whole UK Biobank sample (N = 487,000). EWAS was performed using the –fastGWA-mlm-binary option and filters, keeping SNVs with MAF 5 × 10^−6^ and an SNV messiness rate of 0.02. We included sex, age, batch, and the first ten genetic principal components provided by UK Biobank as covariates in the association analysis to correct for fixed effects. The summary statistics for exome data were calculated on the GRCh38/hg38 genomic build and included 7,890,268 SNVs for CBP and 7,998,110 SNVs for dorsalgia and IDD.

### 2.4. EWAS of Multi-Trait

We used the SHAHER framework, which aims to construct a multi-trait from a group of genetically correlated traits and identify genetic variants controlling this trait [19]. The method suggests that the genetic background of each of the genetically correlated traits can be decomposed into two components: one common to all traits (the shared genetic impact, or SGI) and one specific for each given trait. To find shared genetic variants that explain SGI, a new trait called the shared genetic impact trait (SGIT) is created as a linear combination of the original traits. The coefficients in this combination are defined by maximizing the proportion of SGI in the genetic background of SGIT. To estimate the SGIT coefficients, we utilized the heritability estimates, phenotypic correlation matrices, and genetic correlation matrices assessed beforehand for the original BP-related traits. Then, EWAS summary statistics for SGIT were calculated using the summary statistics calculated for the original traits and the SGIT coefficients. In particular, the z-score for SGIT was calculated as a weighted sum of z-scores for individual traits. The scheme of the SHAHER analysis of the BP-related traits is shown in Figure 1. The SHAHER method has been described by Svishcheva et al. [19]. The current SHAHER analysis is described in detail in the Appendix A.

Pairwise phenotypic correlations between the three original BP traits were assessed in a subsample of non-relatives (307,876 white individuals with CBP, dorsalgia, and IDD status information). Phenotypic correlations of SGIT with CBP, dorsalgia, and IDD were evaluated using summary-level data from imputed genotypes according to the approach proposed by Stephens et al. [21]. We estimated both genetic correlations between the traits and the trait’s SNP-based heritabilities utilizing the LD Score regression tool [22] and summary statistics estimated on imputed variant data. 

### 2.5. Gene-Based Association Analyses

For gene-based association analysis, EWAS summary statistics (z-scores and effect sizes) and the matrices of correlations between genotypes of all SNVs within a gene were used.

#### 2.5.1. Matrices of Genotype Correlations

The correlation between every pair of variants within a gene was estimated using the sequencing genotypes of the 129,807 unrelated white UK Biobank participants (data field 22021) by plink v2.00a3.7LM (https://www.cog-genomics.org/plink/2.0/, accessed on 30 October 2022) with options –mac 3 –geno 0.02.

#### 2.5.2. Variant Annotations

We performed a gene-based association analysis for each of four different variant annotations: LoF, LoF + missense, LoF + protein coding, and all intragenic. Variants were annotated using the Ensembl Variant Effect Predictor (VEP) [23].

The LoF variants include those annotated as frameshift variant, splice acceptor variant, splice donor variant, start loss, stop gain, stop loss, and transcript ablation. The LoF + missense variants additionally include those annotated as transcript amplification, inframe insertion, inframe deletion, missense variant, and protein-altering variant. The LoF + protein coding variants additionally include those annotated as synonymous variant, start retained variant, stop retained variant, coding sequence variant, and incomplete terminal codon variant. The all intragenic variants annotation includes all variants within a gene from 5′UTR to 3′UTR.

For gene-based analysis, we used SNVs matching the following criteria: MAF ≤ 0.01 and minor allele count (MAC) > 10. The genotypes of ultra-rare variants with MAC ≤ 10 were collapsed into a single variant (see below). The numbers of collapsed and other non-ultra-rare (MAC > 10) variants are shown in Appendix A.

Additionally, we used LoF burden tests of association for rare predicted LoF variants [24]. We collapsed all LoF variants within a gene into a single variant using the same technique as for collapsing ultra-rare variants (see below).

#### 2.5.3. Collapsing

For the analysis of ultra-rare variants, we used a technique proposed by Zhou et al. [25]. Before testing each variant annotation, we collapsed the genotypes of variants with MAC ≤ 10 (MAC cutoff details are in Appendix A). In short, the procedure is as follows: individuals with a homozygous rare allele in at least one of the collapsing variants were given a genotype coding of 2, while those with a heterozygous rare allele in at least one of the collapsing variants were given a genotype coding of 1. Individuals with no rare allele in any of the collapsing variants were given a genotype coding of 0. We calculated the summary statistics for collapsed genotypes using the same method as for EWAS and then used the collapsed variants together with all other variants with MAC > 10 for gene-based analysis.

#### 2.5.4. Methods of Gene-Based Analysis

The sumSTAAR framework was used for the gene-based association analysis [26]. Two regression-based methods based on the summary statistics were applied: SKAT-O [27] and PCA [28]. These methods were implemented in the sumFREGAT R-package [29]. The SKAT-O and PCA results were combined using the aggregated Cauchy omnibus test, ACAT-O [30]. The analysis was limited to protein-coding genes with at least two variations that had the EWAS summary statistics.

The Bonferroni adjusted significance and suggestive levels for the total number of genes (20,000) were defined as 2.5 × 10^−6^ and 2.5 × 10^−5^, respectively.

### 2.6. Conditional Analysis

Conditional analysis was performed using the GCTA-COJO tool. [31,32] The COJO selection procedure was applied to SNVs located within 1 Mb of the gene borders. The threshold *p*-value for index variants (–cojo-p) was set as the minimum *p*-value within the gene region being tested. Independent variants selected at this stage were considered conditional if they were located outside of the analyzed gene region. Conditional summary statistics (*p*-values, betas) calculated for the variants within the analyzed gene region were then used as input for gene-based analysis. To take intergenic variants into account, we merged the genotypes and summary statistics of exome data with those of imputed data. CBP summary statistics were calculated for imputed SNVs using the same sample. For SGIT, GWAS summary statistics were calculated on imputed genotypes using the whole sample of white British individuals (N = 449,136).

## 3. Results

### 3.1. Single Variants Association Analysis

We performed EWAS for the three original traits using all genetic variants that passed quality control. The EWAS summary statistics are available at https://mga.icgbio.ru/BP_phens_ewas/ (accessed on 13 October 2023). Manhattan and QQ plots are presented in Appendix A. No significant association signal was detected. Only one variant showed an association at the suggestive level of significance: an ultra-rare variant, 12:14,507,063 (*p*-value = 2.74 × 10^−7^; MAF = 9 × 10^−5^), in the PLBD1 gene for CBP.

To identify genetic variants shared by BP-related traits, we used the SHAHER approach. Using the phenotypic and genetic correlations between the original traits and their SNP-based heritability (Figure 1, first three columns and rows), we estimated the coefficients of an optimal linear combination of the original traits to build SGIT as 0.69, 0.44, and 0.31 for CBP, dorsalgia, and IDD, respectively. Using these coefficients and EWAS summary statistics for the original BP-related traits, we calculated EWAS summary statistics for SGIT. Manhattan and QQ plots are presented in Appendix A. There were three variants significantly associated with SGIT and ten variants suggestively associated with SGIT. However, none of these variants was included in the gene-based analysis because some of them were located outside the genes, while others had MAC ≤ 10 and collapsed with other ultra-rare variants.

Then, we calculated the phenotypic and genetic correlations between the original traits and SGIT and estimated the SNP-based heritability of SGIT (Figure 1, fourth column and row). As can be seen, the phenotypic correlations between SGIT and the original traits are stronger than those between the original traits. All genetic correlations between SGIT and the original traits are no less than 0.93. The SNP-based heritability of SGIT is higher than that of original traits.

### 3.2. Gene-Based Association Analysis

We formed the four different variant annotations (LoF, LoF + missense, LoF + protein coding, and all intragenic) within every gene. Then, we collapsed variants with MAC ≤ 10 and calculated their summary statistics. Association results for collapsed variants are freely available in the ZENODO database (https://doi.org/10.5281/zenodo.8118630, accessed on 31 March 2023). No significant or suggestive association signals were detected for the collapsed variants.

The results of gene-based association analysis with *p*-value ≤ 2.5 × 10^−5^ are shown in Table 1. Full results are available in the ZENODO database (https://doi.org/10.5281/zenodo.8118630, accessed on 31 March 2023). QQ plots for gene-based analyses are shown in Appendix A.

Only one of the three original traits, namely CBP, showed a significant gene-based association with two genes: *P4HTM* (LoF + protein coding) and *SLC13A1* (LoF + missense). *P4HTM* also showed a suggestive association signal for CBP on another variant annotation (LoF + missense). A few genes provided associations at the suggestive significance level: *KIF5B* and *TIMM44* for CBP; *RPL37* and *AVPR1A* for dorsalgia; and *RNF15*, *KIAA2012*, and *LETMD1* for IDD. For the LoF burdens, there was no association with a *p*-value ≤ 2.5 × 10^−5^.

The gene-based association analysis of SGIT detected a significant association for the *FSCN3* gene (LoF annotation) (Table 1). Table 2 shows the *p*-values estimated for this gene on SGIT and the original traits. As can be seen, these values are low for all traits, especially for the LoF annotation.

We also performed the gene-based association analysis of SGIT using the LoF burden test. No significant association was found. The *IL1R2* gene showed an association at the suggestive level (*p*-value = 1.14 × 10^-−5^, Table 1). The *p*-values estimated for this gene and the original traits were 6.4 × 10^−5^, 0.09, and 0.004 for CBP, dorsalgia, and IDD, respectively.

### 3.3. Conditional Analysis

We performed conditional gene-based association analysis for *SLC13A1* (CBP, LoF + missense), *P4HTM* (CBP, LoF + protein coding), and *FSCN3* (SGIT, LoF). *SLC13A1* and *FSCN3* harbored SNVs with the lowest *p*-values within 1 Mb from their borders; thus, no conditional SNVs were selected, and the gene-based *p*-values remained unchanged. For *P4HTM*, the strongest signal within the gene (*p*-value = 0.0037) was in high LD with an imputed SNV with the *p*-value = 0.0009. As a result, the conditional gene-based *p*-value for *P4HTM* was 0.001.

Next, we checked the influence of the genome-wide significant SNVs already known from prior studies for CBP and related traits within 1 Mb from gene borders on the association of *SLC13A1*, *P4HTM*, and *FSCN3*. We did not find such genome-wide significant SNVs for *FSCN3*. The *SLC13A1* gene contained only one genome-wide significant SNV, rs28364172 [8]; no such variants were outside the gene. It has been shown that multiple significant GWAS signals are located around the *P4HTM* gene: rs34762726 [8], rs1491985 [33], and 14 SNVs found in the Finngen study for pain, lower back pain, dorsalgia, and other dorsopathies (https://r8.finngen.fi/, accessed on 10 December 2022). Of these 16 SNVs, 10 were present in our sample. However, their r^2^ with the top two SNVs within the *P4HTM* gene were rather small and did not exceed 0.02. This was not surprising, as most of the GWAS signals were obtained on common variants. Being used as conditional SNVs, they did not affect the significance of *P4HTM* gene-based statistics.

## 4. Discussion

We performed exome-wide gene-based analysis using the UK biobank data and detected two genes, *SLC13A1* and *FSCN3*, that were significantly associated with BP-related traits. The first gene was associated with CBP, and the second gene was associated with SGIT. 

The *SLC13A1* gene has been previously detected as being associated with BP-related traits due to LoF variants [8,18]. This gene encodes a protein that functions as a high-affinity sodium-dependent sulfate transmembrane transporter [34]. A deficiency of the SLC13A1 protein is associated with a reduced blood sulfate level, which plays a key role in the underlying processes leading to painful IDD [8].

To increase the power of the association analysis, we applied the new approach based on the following. A modern model of back pain suggests that this condition is influenced by a complex interplay of biological, psychological, and social factors [11,16,35,36]. The lack of a standard definition of BP poses a significant challenge in identifying new genetic loci and determining whether the identified variants contribute to the risk of underlying spine pathology or the development of persistent pain (chronicity). To address these challenges and enhance our understanding of the genetic influence on BP, we used an innovative strategy. We considered three different back pain-related phenotypes: one self-reported and two defined by EHR-based codes. EHR-based back pain-related phenotypes such as dorsalgia and IDD differ from self-reported CBP in that they reflect back pain of a severity sufficient to warrant health care seeking and are derived from longitudinal health care records, often over the span of many years, rather than point prevalence from a single survey. Nevertheless, the genetic nature of all these phenotypes is common because the genetic correlations between them are about 90%. We explored shared pathways among different pain phenotypes using multi-trait methodologies, which can help reduce heterogeneity [6,19]. 

Due to this approach, we identified the second gene, *FSCN3*, which is a new gene associated with BP-related traits. This gene was identified using LoF variants, making *FSCN3* a highly intriguing potential drug target. The association of a gene with disease through LoF variants presents several advantages as a drug target. Firstly, LoF variants often lead to a complete or partial loss of gene function, which can directly and measurably impact disease. This clear functional connection allows for the development of targeted therapies to restore or replace the lost function. Secondly, since LoF variants typically affect a specific gene, drugs can be designed to precisely target the gene or its product, enhancing efficacy while minimizing off-target effects or side effects. Additionally, gene therapy or other genetic interventions can be employed to repair or replace the faulty gene, offering a potential long-term solution to the problem. Therefore, the association of *FSCN3* with BP-related traits through LoF variants provides a clear functional link, target specificity, potential for genetic rescue, predictive value, and personalized medicine opportunities, making it a promising drug target for therapeutic development.

So far, little is known about the functions of *FSCN3*. *FSCN3* encodes the fascin actin-bundling protein 3. This gene is predicted to enable actin filament binding activity and to be involved in actin filament bundle assembly, cell migration, and establishment or maintenance of cell polarity. It is predicted to be located in the cytoskeleton and to be active in several cellular components, including lamellipodium, microvillus, and ruffle (provided by the Alliance of Genome Resources, April 2022 (www.alliancegenome.org, accessed on 15 December 2022)). A related protein, Fascin 1, was shown to contribute to neuropathic pain by promoting inflammation in rat spinal cord [37]. It is highly expressed specifically in microglia after spinal cord injury and regulates microglial migration in mice [38]. *FSCN3* is also involved in neurogenic pain. It showed decreased expression in the model of neuropathic pain ‘spare nerve injury’ in rats (*p*-value = 0.007, (https://maayanlab.cloud/Harmonizome/gene_set/Neurological+pain+disorder_Dorsal+Root+Ganglia_GSE15041/GEO+Signatures+of+Differentially+Expressed+Genes+for+Diseases, accessed on 15 December 2022) [39]). It is also possible that *FSCN3* is involved in other biological processes because the FSCN3 protein interacts with several other proteins, including FLAD1, GLMN, PPP5C, and PRAME (www.alliancegenome.org, accessed on 15 December 2022). One of them, GLMN, is involved in the inflammation process via the regulation of C-reactive protein [40]. We can speculate that the loss of FSCN3 function changes the protein complex and is involved in the inflammation process. Inflammation is often accompanied by pain, and due to it, the loss of FSCN3 function could theoretically explain back pain.

We did not have the ability to replicate *FSCN3* because its LoF variants are ultra-rare (MAF ~ 10^−6^–10^−4^); they are absent in the list of GWAS variants and may be replicated only on exome sequencing data. Nevertheless, there are three suggestively significant (*p*-value < 2.5 × 10^−7^) GWAS signals found in the FinnGen study for pain (limb, back, neck, head abdominally) within ~250 Kb from *FSCN3* borders (rs138017935, rs185233392, and rs147079677, (https://r8.finngen.fi/, accessed on 10 December 2022). The latter two SNVs also have *p*-value = 5 × 10^−6^ for dorsalgia. This supports the hypothesis of *FSCN3* involvement in pain development. The replication of more frequent LoF variants is also very difficult. For example, the LoF variants with MAF ~ 10^−3^ in the known *SLC13A3* gene can not be replicated using FinnGen because their frequencies in the Finnish population are lower than in the UK Biobank, and these variants were not included in the FinnGen GWAS. Even without being replicated, the *FSCN3* gene deserves attention because it is only the second gene shown to have an effect on back pain due to LoF variants.

We also detected several genes associated with BP-related traits with a suggestive level of significance. The *IL1R5* gene was associated with SGIT (*p*-value = 1.14 × 10^−5^ for the LoF Burden test). For this gene, an association with osteoporosis has been shown in the Chinese population [41]. Osteoporosis is a common metabolic bone disease characterized by progressive bone mass loss and the degeneration of bone microarchitecture [42]. *IL1R2* has also been shown to be associated with ankylosing spondylitis [43]. This disease is characterized by inflammation of the spine and sacroiliac joints, causing pain and stiffness and, ultimately, new bone formation and progressive joint ankylosis.

The *KIF5B* gene was associated with CBP (*p*-value = 1.94 × 10^−5^ for LoF + protein coding variants). Three pathogenic variants in this gene have been described as associated with kyphomelic dysplasia (MIM: 211350), which belongs to a group of heterogeneous skeletal dysplasias [44]. Therefore, both genes *IL1R2* and *KIF5B* are good candidates for explaining some cases of back pain.

One more gene associated with CBP is *P4HTM*. A minimum *p*-value of 2.9 × 10^−7^ was obtained for the “LoF + protein coding” variant annotation. The protein coded by *P4HTM* belongs to the prolyl hydroxylase domain (PHD). The PHD subfamily has four members: PHD1, PHD2, PHD3, and PHD4, or P4HTM [45]. The PHD proteins play a critical role in the regulation of hypoxia-inducible transcription factors (HIF) under normoxia. HIF plays a role in adaptation to hypoxia and may be related to cellular oxygen sensing. Increasing evidence suggests that the HIF pathway plays an essential role in bone regeneration and repair [46] because the initial factor of bone regeneration is the upregulation of HIF-1α induced by hypoxia [47]. We can speculate that the abnormal structure of the P4HTM protein caused by rare mutations impairs bone regeneration and thus causes back pain. It is also known that pathogenic variants of *P4HTM* in homozygous or compound heterozygous states are the cause of the HIDEA syndrome (OMIM # 618493) [48]. Among other symptoms, the phenotype is characterized by muscular hypotonia [49]. We can speculate that one more possible mechanism of *P4HTM* involvement in back pain is muscular hypotonia resulting from mutations in *P4HTM*. However, this gene did not pass the conditional analysis. The conditional SNV that affected the *P4HTM* signal was not genome-wide significant or suggestive (*p*-value = 0.0009) and was neither present among the known CBP variants nor in LD with them. It is located in the intron of *MST1R*, a macrophage-stimulating 1 receptor gene, yet it showed a *p*-value lower than the exonic *P4HTM* SNV in LD with it. Given this, we cannot fully claim *P4HTM* to be associated with CBP, nor can we reject that it may have a potential role in CBP pathogenesis. The region is known to be associated with pain phenotypes, such as IDD, dorsalgia, pain (limb, back, neck, head abdominally), lower back pain or/and sciatica, hernia, and other dorsopathies, with 16 genome-wide significant SNVs already found within 1 Mb from *P4HTM* borders [8,33] (https://r8.finngen.fi/, accessed on 10 December 2022). The region is also dense in genes: 16 SNVs are located in 10 genes. Further studies are needed to prioritize the gene(s) in this location that actually affect the back pain phenotype.

## 5. Conclusions

We conducted the exome-wide gene-based association analysis of BP-related phenotypes using exome sequencing data from a UK biobank project. We identified two genes significantly associated with BP-related phenotypes. The effect of the known *SLC13A1* gene on SBP was explained not only by LoF but also by rare missense variants. A newly detected gene, *FSCN3*, is a promising potential drug target because its effect on BP is explained by LoF variants. The role of this gene should be clarified in further studies.

## Data Availability

All data are freely available in the ZENODO database https://doi.org/10.5281/zenodo.8118630 (accessed on 31 March 2023) and at https://mga.icgbio.ru/BP_phens_ewas/ (accessed on 13 October 2023).

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
