# Peer review of "Multi-Trait Exome-Wide Association Study of Back Pain-Related Phenotypes"

_genes, 2023, doi:10.3390/genes14101962_

Round 1
Reviewer 1 Report
Some of the co-authors of this article developed a method for combining multiple phenotypes. In this article the authors have applied the previously developed method to derive a combined pain phenotype and performed EWAS on this trait. This article is very well written.
The authors may consider the comments for revision.
A sample characteristic table with summary statistics on age, sex and CBP, dorsalgia and IDD and the overlap among these traits would have been useful. It is not clear if the correlation measures (any measure on contingency table, phi for example) based on the raw data would be similar to the correlations derived based on the EWAS summary statistics. How was the SGIT, derived as a linear combination of the pain phenotypes, treated (continuous or categorical variable) for EWAS? There may be only categories of values for this variable and hence the histogram of this data may be useful. Is this variable significantly different among the original pain phenotypes? The summary table may also include this information.
The EWAS method used for SGIT should be mentioned in the methods section. Since SGIT is a linear combination derived based on the genetic information from the other 3 traits, it may not be surprising if some additional EWAS hits for this variable is found. How SGIT would compare with the total pain score of the three pain measures (though it is not optimised for genetic correlations) which can be a useful measure of amount of pain. Also, it is possible to drive a combined phenotype of the pain measures, for example, all 3 pains or at least one or two pains etc. The controls for such analysis could be based participants with none of the pain measures.
It would be very useful to provide the link for the summary statistics of individual EWAS.
Perception of pain among the sexes may be different and there is enough literature on this topic. A stratified analysis for each sex may provide some additional insight into the genetics of pain, as the sample size in UK biobank would be sufficient for such an analysis.
Reviewer 2 Report
In this manuscript, the authors performed exome-wide association analysis and gene-based analysis using the UK biobank data to identify genes that are significantly associated with back-pain-related traits. The structure of the manuscript is complete.
Major comments:
1. It would be better to summarize the recent genetic findings of the back pain in the introduction part. “Several dozen loci associated with CBP, dorsalgia, and IDD have been identified by genome-wide association studies” seems to be too vague.
2. Multivariate association analysis is a confusing term to describe the SHAHER framework.
3. The idea of finding the optimal linear combination of the original traits to build SGIT is not a common approach. There are other multi-trait analysis methods, e.g. Multi-trait analysis of genome-wide association summary statistics using MTAG from Nature Genetics. Did you try that method before?
4. For the SHAHER results, none of them are included in the following analysis. Did you validate those signals identified?
5. Which p-value threshold are you using? The table 1 is using 2.5 * 10^-5 while in the method, “The Bonferroni adjusted significance level for the total number of genes (20 000) was defined as 2.5×10^-6”
6. For the gene-based analysis, did you compare the results of sumSTAAR with other gene-based methods? It would be more convincing if you include other association test results, e.g. SKAT to confirm that the signal is not false positive.
Minor editing is needed.
